# OpenReview forum: "CRITIC: Large Language Models Can Self-Correct with Tool-Interactive Critiquing"
_ICLR.cc/2024/Conference — ICLR 2024 poster_

### Official Review · Reviewer_hutJ · 2023-10-31

**Soundness:** 3 good
**Presentation:** 3 good
**Contribution:** 2 fair
**Rating:** 5
**Confidence:** 3

**Summary:**

The papers proposes CRITIC, a framework for composing programs involving LMs self-correcting themselves using external tools. The authors conduct experiments with question answering, program synthesis and toxicity reduction show that CRITIC consistently improves the performance of LLMs.

**Strengths:**

1. The paper is written clearly and easy to follow.
2. I found the comparison with ReAct interesting, i.e. the role of parameter knowledge vs language feedback.

**Weaknesses:**

1. I'm not convinced the CRITIC framework is novel enough to count as a contribution. The idea idea of using natural language feedback [1, 2, 3, 4] that guides LMs in revising their responses is pretty old as is the idea of using tools [5,6]. I agree the authors provide a nice unifying framework and some new downstream tasks (e.g. toxicity with PerspectiveAPI), but these don't seem to be pass the bar for ICLR.

2. The authors don't compare with other frameworks endowing LMs with self-correction and tool use, like the ones listed above.

3. I think the claim that tool use "mimic[s] human thinking and behavior" is overblown. Humans use think and work with tools very differently, typically not through a text-only interface.

[1] https://arxiv.org/abs/2204.14146

[2] https://arxiv.org/abs/2303.11366

[3] https://arxiv.org/abs/2212.08073

[4] https://arxiv.org/abs/2303.16749

[5] https://arxiv.org/abs/2207.14502

[6] https://arxiv.org/abs/2302.04761

[7] https://openai.com/blog/function-calling-and-other-api-updates

**Questions:**

How does the paper compare with other frameworks endowing LMs with self-correction and tool use?

---

> ### Author Response · Authors · 2023-11-13
> **Author Response to Official Review by Reviewer hutJ (1/2)**
>
> Thank you for thoroughly reviewing our work! We have carefully considered all your concerns and addressed them in the following comments.
>
> ## CRITIC Compared to Existing Works on Self-Correction and Tool-use
>
> > The idea idea of using natural language feedback [1, 2, 3, 4] that guides LMs in revising their responses is pretty old as is the idea of using tools [5,6]. I agree the authors provide a nice unifying framework and some new downstream tasks (e.g. toxicity with PerspectiveAPI), but these don't seem to be pass the bar for ICLR.
>
> Thank you for raising these concerns! Here we discuss why we believe CRITIC is novel, and more than a trivial blend of existing works on self-correction [1, 2, 3] and tool use [4,5]. While these are crucial fast-moving areas of LLMs, CRITIC offers unique findings, integrated frameworks, and valuable insights, making CRITIC a distinct contribution.
>
> The relevant papers in these two fields you mentioned actually **have very different research perspectives and stances from CRITIC**:
>
> ### **Intrinsic Self-Correct with NL feedback**
>
> Works include Self-Critique [6], CAI [7], Reflexion[1], Self-Refine [2], and others [8-10] you mentioned, they prompt or train language models to correct their results. In contrast, our study is the first to demonstrate that such a "Self-Verification and Self-Correction" approach has proven to be remarkably unreliable across diverse tasks and various LLMs (Sec. 4 and Appendix D.1). Specifically, **modest improvements or even deterioration are observed universally using self-correct without external feedback**. Consequently, CRITIC emphasizes the importance of feedback from external interactions for the consistent self-improvement of LLMs. The proposed framework is general and has proven effective across multiple tasks. These profound reflections on unreliable self-correction of LLMs and crucial findings on the importance of external feedback can be important learnings for the community, as recognized by Reviewer [rWYL].
>
> ### **The Unreliability of Self-correction**
>
> Moreover, CRITIC further delves into the core reason behind the unreliability of self-verification from the perspective of uncertainty estimation, as shown in Appendix D.1. Essentially, the models are **incapable of accurately identifying "what they know" (i.e., LLMs don't know what they know)** [12] without relying on external tools. Therefore, without the aid of oracle verification (employed in many contemporary works such as Reflexion [1] and Self-Refine [2]), self-correction might surprisingly deteriorate performance for many tasks, even worsening the initial answer (as demonstrated in Table 2, 3 under CRITIC w/o Tool, and in Table 8 under Self-Refine).
>
> As noted by Reviewer [dDfr], a nice recent follow-up work to CRITIC, titled "LLMs cannot self-correct reasoning yet" [14] extends the study of CRITIC (which cites CRITIC numerous times). It further validates and expands our findings on the unreliability of Self-Verify and Self-Correct in CRITIC using GPT-4 in more settings on many reasoning tasks.
>
> ### **Tool-Use**
>
> Another category of related work is Tools Augmented LMs, such as ReAct [4], PoT [5], and Toolformer [17]. We would like to clarify that these works significantly differ from CRITIC as they focus on tool learning. To the best of our knowledge, **none of them consider using tool-interaction as faithful feedback to iteratively improve the model, which is the key component of CRITIC.**
>
> In conclusion, CRITIC is the first to unveil the unreliability of self-verification and self-correction across diverse tasks and LLMs of various families and sizes. By initially highlighting the challenges LLMs encounter in self-verification [12], self-correction [6,7], our goal is to rectify any potential overestimations of these LLM's abilities within the research community [1-3]. By emphasizing that feedback from external tool interaction is crucial for consistent self-improvement of LLMs, we hope our findings provide valuable insights and encourage further exploration and enhancement of self-improving LLMs.

---

> ### Author Response · Authors · 2023-11-13
> **Author Response to Official Review by Reviewer hutJ (2/2)**
>
> ## Experiments of CRITIC Comparing Other Self-Correct and Tool-Use Frameworks
>
> > The authors don't compare with other frameworks endowing LMs with self-correction and tool use, like the ones listed above.
>
> In fact, CRITIC has been compared with the most advanced baselines of Self-Correct and Tool Use across multiple tasks in the original paper:
>
> - Firstly, the work you mentioned includes methods for self-correction based on natural language feedback [1-3, 6-10], which is akin to the "CRITIC w/o Tool" baseline implemented and compared in many tasks in our paper, which relies on self-correction without using external tools.
> - Moreover, in the QA task, CRITIC is even compared with a concurrent work, Self-Refine [2], as shown in Table 5. We also compared it with the Self-Correct [11] method that trains an additional corrector in the Toxicity Reduction task. We have conducted detailed comparisons and discussions on related baselines in all experiments and in Appendix C.1.
> - Additionally, you mentioned the tool-use works like Toolformer [17] and many others [3,4]. We have actually compared the most advanced tool-use methods in each domain, such as the search engine-based ReAct [4] in the QA tasks; PoT [3] that used a Code Interpreter in the Math Program Synthesis tasks; Quark [18] and Self-Correct [11] that utilized Perspective API for Toxicity Reduction, etc.
>
> We have ensured fairness in tool use and self-correct settings for all baselines across all these comparative experiments covering a wide diversity of tasks. These comprehensive comparisons across various tasks, LLMs, and model sizes collectively demonstrate the generality and effectiveness of CRITIC.
>
> ## Revising the phrase “allows LLMs to mimic human thinking and behavior”
>
> > I think the claim that tool use "mimic[s] human thinking and behavior" is overblown. Humans use think and work with tools very differently, typically not through a text-only interface.
>
> We sincerely appreciate your feedback regarding the wording used here!
>
> Our intended message is that CRITIC is a framework designed, inspired by the process of human interaction with tools, such as humans compiling and running their own programs, receiving feedback, reflecting on issues, and going through the debugging process. We also agree that humans may exhibit a much richer inner monologue and behavior in using tools and reflecting, and language is indeed a limited interface through which LLM interacts with the world. In response to your feedback, we will revise this phrase to "allows for human-like verify-then-correct trajectories", and we hope this address your concerns.
>
> In light of these clarifications and revisions, we kindly request that you consider increasing the review score.

---

> ### Author Response · Authors · 2023-11-13
> **Author Response to Official Review by Reviewer hutJ (References)**
>
> ### References
>
> [1] Shinn, Noah, et al. "Reflexion: Language agents with verbal reinforcement learning." Thirty-seventh Conference on Neural Information Processing Systems. 2023.
>
> [2] Madaan, Aman, et al. "Self-refine: Iterative refinement with self-feedback." Thirty-seventh Conference on Neural Information Processing Systems. 2023.
>
> [3] Chen, Xinyun, et al. "Teaching large language models to self-debug." arXiv preprint arXiv:2304.05128 (2023).
>
> [4] Yao, Shunyu, et al. "React: Synergizing reasoning and acting in language models." arXiv preprint arXiv:2210.03629 (2022).
>
> [5] Chen, Wenhu, et al. "Program of thoughts prompting: Disentangling computation from reasoning for numerical reasoning tasks." arXiv preprint arXiv:2211.12588 (2022).
>
> [6] Saunders, William, et al. "Self-critiquing models for assisting human evaluators." arXiv preprint arXiv:2206.05802 (2022).
>
> [7] Bai, Yuntao, et al. "Constitutional ai: Harmlessness from ai feedback." arXiv preprint arXiv:2212.08073 (2022).
>
> [8] Campos, Jon Ander, and Jun Shern. "Training language models with language feedback." ACL Workshop on Learning with Natural Language Supervision. 2022.. 2022.
>
> [9] Chen, Angelica, et al. "Improving code generation by training with natural language feedback." arXiv preprint arXiv:2303.16749 (2023).
>
> [10] Haluptzok, Patrick, Matthew Bowers, and Adam Tauman Kalai. "Language models can teach themselves to program better." *arXiv preprint arXiv:2207.14502* (2022).
>
> [11] Welleck, Sean, et al. "Generating Sequences by Learning to Self-Correct." *The Eleventh International Conference on Learning Representations*. 2023.
>
> [12] Kadavath, Saurav, et al. "Language models (mostly) know what they know." arXiv preprint arXiv:2207.05221 (2022).
>
> [13] Ni, Ansong, et al. "Lever: Learning to verify language-to-code generation with execution." International Conference on Machine Learning. PMLR, 2023.
>
> [14] Huang, Jie, et al. "Large language models cannot self-correct reasoning yet." arXiv preprint arXiv:2310.01798 (2023).
>
> [15] Stechly, Kaya, Matthew Marquez, and Subbarao Kambhampati. "GPT-4 Doesn't Know It's Wrong: An Analysis of Iterative Prompting for Reasoning Problems." arXiv preprint arXiv:2310.12397 (2023).
>
> [16] Valmeekam, Karthik, Matthew Marquez, and Subbarao Kambhampati. "Can Large Language Models Really Improve by Self-critiquing Their Own Plans?." arXiv preprint arXiv:2310.08118 (2023).
>
> [17] Schick, Timo, et al. "Toolformer: Language models can teach themselves to use tools." arXiv preprint arXiv:2302.04761 (2023).
>
> [18] Lu, Ximing, et al. "Quark: Controllable text generation with reinforced unlearning." Advances in neural information processing systems 35 (2022): 27591-27609.

---

> ### Comment · Reviewer_hutJ · 2023-11-21
> **Thanks for the response**
>
> Thanks for the detailed response, I appreciate engaging with all the points I raised.
>
> However, I remain somewhat unconvinced that using external feedback counts as a novelty. The idea of using external feedback in combination with language feedback was used before (e.g. [1, 2]). Moreover, the need to rely on external tools is generally a limitation: it means that LLM capabilities cannot scale up unless new tools are available. It seems that at least some method can overcome it in certain settings. I can imagine a paper focusing on this limitation (similar to [3]), but that would require a change of the story in the paper.
>
> [1] https://arxiv.org/abs/2303.16749
>
> [2] https://arxiv.org/abs/2303.11366
>
> [3] https://arxiv.org/abs/2310.01798

---

> > ### Author Response · Authors · 2023-11-21
> > **Thank you for your reply!**
> >
> > Thank you for your additional comment and for taking the time to engage with us on this matter! We appreciate your perspective and understand that you remain unconvinced about the novelty of using external feedback. However, we would like to continue addressing your concerns and clarify the unique contributions of CRITIC.
> >
> > ### **The novelty of using tool-interaction for feedback**
> >
> > > The idea of using external feedback in combination with language feedback was used before (e.g. [1, 2]).
> > >
> >
> > We agree the concept of using external feedback in combination with language feedback has been used in prior studies. However, CRITIC's approach to this feedback is distinct and novel. The cited works [1, 2] utilize task completion oracle information (i.e., ground truth) as external feedback. In contrast, CRITIC empowers LLMs to verify their output's truthfulness, feasibility, and safety through tool-interaction without relying on task oracle feedback. For instance, by accessing web knowledge via Google to validate their internal knowledge and make further corrections. Additionally, CRITIC firstly highlights the limitations of self-correction and proposes an integrated framework for self-correction based on LLM’s automatic external tool interaction. This comprehensive and practical approach, alongside the explicit study of self-correction limitations, represents a novel contribution to the field.
> >
> > ### **The necessity and advantage of using external tools**
> >
> > > Moreover, the need to rely on external tools is generally a limitation: it means that LLM capabilities cannot scale up unless new tools are available.
> > >
> >
> > We also agree that the requirement for external tools can be viewed as a limitation, as it indeed signifies that LLM capabilities may not scale on new tasks without new tools. However, this also realistically reflects the current state and limitations of LLMs, which in turn motivated us to introduce the unified and general CRITIC framework for integrating feedback from tool-interaction. As a result, the utilization of external tools can be advantageous. It allows LLMs to leverage the continuous evolution of toolsets and resources for reliable self-improvement, often encapsulating domain-specific expertise and data not inherently available to LLMs.
> >
> > > It seems that at least some method can overcome it in certain settings.
> > >
> >
> > We believe that the necessity of external tools is not something that can be easily overcome in certain settings, even for the strongest LLMs. Let's consider an extreme scenario where a “future method” enables GPT-10 to robustly handle all scenarios and generate correct facts with a 99% accuracy rate. However, due to the potential risks associated with the 1% error rate, humans still cannot trust the results and achieve trustworthy automation. In such cases, we believe that the CRITIC framework, which drives AI to autonomously use external tools or interact with the environment for self-verification and correction, is still necessary. This might even be a key pathway to achieve the “Scalable Oversight” vision [4].
> >
> >
> > ### **The focus of the paper**
> >
> > > I can imagine a paper focusing on this limitation (similar to [3]), but that would require a change of the story in the paper.
> > >
> >
> > We appreciate your suggestion to change the paper's focus to the limitations of LLMs without external tools, similar to the cited follow-up work of CRITIC [3]. However, we believe that the current focus of the paper, which highlights the limitations of self-correction and proposes a novel framework to combine self-correction and external tool interaction, is also a valuable contribution to the field. And we have made a concerted effort to clearly communicate this focus throughout the paper.
> >
> > In light of these clarifications, we kindly request that you reconsider the novelty and contributions of CRITIC. We are also open to further discussions and feedback to improve our work.
> >
> > ---
> >
> > ### References
> >
> > [1] Chen, Angelica, et al. "Improving code generation by training with natural language feedback." arXiv preprint arXiv:2303.16749 (2023). https://arxiv.org/pdf/2303.16749.pdf
> >
> > [2] Shinn, Noah, et al. "Reflexion: Language agents with verbal reinforcement learning." Thirty-seventh Conference on Neural Information Processing Systems. 2023. https://arxiv.org/pdf/2303.11366.pdf
> >
> > [3] Huang, Jie, et al. "Large language models cannot self-correct reasoning yet." arXiv preprint arXiv:2310.01798 (2023). https://openreview.net/forum?id=IkmD3fKBPQ
> >
> > [4] Bowman, Samuel R., et al. "Measuring progress on scalable oversight for large language models." arXiv preprint arXiv:2211.03540 (2022). https://arxiv.org/pdf/2211.03540

---

> > > ### Comment · Reviewer_hutJ · 2023-11-22
> > > **Thanks for the response**
> > >
> > > Thanks for the response. After carefully reading our discussion as well as discussions with other reviewers I decided to keep my score.

---

> > > > ### Author Response · Authors · 2023-11-22
> > > > **Thank you for your time!**
> > > >
> > > > Dear Reviewer hutJ,
> > > >
> > > > We deeply appreciate the time and effort you've dedicated to reviewing our paper! In the spirit of the season, we wish you a Happy Thanksgiving!
> > > >
> > > > Best Regards,
> > > > The Authors

---

### Official Review · Reviewer_rWYL · 2023-11-01

**Soundness:** 3 good
**Presentation:** 4 excellent
**Contribution:** 3 good
**Rating:** 8
**Confidence:** 5

**Summary:**

This paper proposes a framework called CRITIC to progressively validate and revise the output based on the feedback from tools. Six different external tools are used including Knowledge base, code interpreter, Text APIs, Wiki, Calculator and Search Engine. Evaluations are done on free-form question answering, mathematical program synthesis, and toxicity reduction. CRITIC was shown to have superior performance on these benchmarks compared to strong baselines including CoT, Self-Consistency, ReAct, and PoT.

**Strengths:**

1. LLM Tool Use is a very timely research topic, and it is an important research area to use external feedback for the self-improvement of LLMs. This paper covers a wider range of tools compared to many prior works which typically employ one single type of tools.
2. The results are rather strong with universal improvements across most tasks evaluated with several different model families and sizes.
3. The ablation against CRITIC w/o Tool shows the importance of external feedback from Tools, which is an important learning for the community.
4. The paper is very well written and is easy to understand with comprehensive comparisons to strong baselines.

**Weaknesses:**

Error analysis is missing on what are the failure modes after using Tools for feedback.

**Questions:**

1. It is unclear how important each Tool is to each task. Such analysis will provide further insight into where the improvements come from.
2. The authors used different sampling config for the experiments for different tasks: e.g. p=0.9 was used for section 4.3 which is different from p=0.5 in 4.1 and 4.2.

---

> ### Author Response · Authors · 2023-11-13
> **Author Response to Official Review by Reviewer rWYL (1/2)**
>
> Thank you for your thorough review! We appreciate your acknowledgment of CRITIC's importance, clarity, novelty, effectiveness, and consistent performance gains. We're grateful for your recommendation to accept our paper.
>
> ## Adding Detailed Error Analysis on QA and Mathematical Program Synthesis Tasks
>
> > Error analysis is missing on what are the failure modes after using Tools for feedback.
>
> Thank you for your constructive suggestion! Here, we point out that the appendix of the original paper actually includes extensive case studies on failure modes of all tasks.
>
> Moreover, in light of your suggestion, we have included new Error Analyses for the QA and Mathematical Program Synthesis tasks, as illustrated below.
>
> ### **1. Success and Failure Case Studies**
>
> Due to the limitation of space in the main text, we provide a representative success and failure case analysis for each task in Appendix E in the original paper, where typical failure modes include evidence search failure, reasoning errors for QA tasks, and verification or correction errors in GSM8k.
>
> ### **2. Error Analysis on QA Tasks**
>
> Following your suggestion, in order to further understand the failure modes after using tools for feedback, we randomly selected 100 cases from the HotpotQA task, and manually annotated and analyzed the error types for both the initial CoT and CRITIC. The results are as follows:
>
> | Error Type                               | Explanations                                                             | CoT | CRITIC |
> |------------------------------------------|--------------------------------------------------------------------------|-----|--------|
> | Hallucination                            | Hallucination, including misinterpreting evidence or inconsistencies | 36% | 7%     |
> | Reasoning Error                          | Incorrect logical reasoning                                              | 5%  | 10%    |
> | Irrelevant Response                      | Answering a question that was not asked                                  | 9%  | 7%     |
> | Insufficient Evidence (Refusal)          | Refusal to answer the question due to lack of sufficient evidence    | 2%  | 12%    |
> | Undefined Answer                         | Providing an empty answer or failing to derive an answer                 | 18% | 5%     |
> | Incorrect Correction                     | CRITIC wrongly altered the correct initial CoT answer                    | -   | 10%    |
> | Label Ambiguity (FN)                     | The prediction is correct but not matching the label                     | 20% | 37%    |
> | Incorrect Label (FN)                     | The dataset answer is incorrectly labeled                                | 9%  | 10%    |
> | Outdated Label (FN)                      | The dataset answer label is outdated                                     | 0%  | 2%     |
>
> As depicted in the table:
>
> - (1) CRITIC can significantly reduce hallucinations (36% vs. 7%), but not all of them. Even after utilizing CRITIC, hallucinations persist due to the inability to find useful evidence via a search engine or misunderstanding the evidence. This is illustrated in Appendix E.
> - (2) Most errors after applying CRITIC arise from reasoning mistakes, refusal to answer, and incorrect corrections. The refusal to answer occurs when CRITIC can't find enough evidence to support a response, which we consider an expected behavior to maintain truthfulness.
> - (3) In reality, CRITIC has effectively helped us identify a large number of label ambiguities, label errors, and outdated issues in the HotpotQA dataset (49% in CRITIC error samples). These false negatives (FN) indicate a certain bias in evaluating different methods on free-form QA tasks using automatic metrics like EM / F1. This has motivated subsequent research to design a more reliable LLM-based evaluation for QA tasks [1].

---

> ### Author Response · Authors · 2023-11-13
> **Author Response to Official Review by Reviewer rWYL (2/2)**
>
> ### **3. Error Analysis on Mathematical Program Synthesis tasks**
>
> On Mathematical Program Synthesis tasks, to offer readers a more comprehensive understanding of the specific corrections made by CRITIC and the specific benefits derived from tool feedback (also mentioned by reviewer ASEM), we carried out a manual statistical analysis of the types of corrections made by CRITIC on the GSM8k full test set (1319 samples).
>
> Specifically, we identified four different categories of initial program errors: syntax errors, runtime errors, unreasonable outputs (such as irrational negative values), and other intrinsic reasoning errors. We calculated the accuracy of the initial PoT (Init), and CRITIC for each type of error. The settings for corrections are consistent with the non-oracle setting in the original paper, with up to four rounds of correction. The statistics are presented in the following table:
>
> | Error Type          | Init (Count) | Init (Acc) | CRITIC (Count)| CRITIC (Acc)|
> |---------------------|-------------|----------|-------------|------|
> | Intrinsic Error     | 281 (77.4%) | 0.0      | 206 (71.8%) | 26.7 |
> | Unreasonable Output | 61 (16.8%)  | 0.0      | 26 (9.1%)   | 57.4 |
> | Syntax Error        | 17 (4.7%)   | 0.0      | 11 (3.8%)   | 35.3 |
> | Runtime Error       | 4 (1.1%)    | 0.0      | 3  (1.0%)   | 25.0 |
> | All Init Errors     | 363         | 0.0      | 246 (85.7%) | 32.2 |
> | Wrong Correction    | -           | 100.0    | 41 (14.3%)  | 95.7 |
>
> As can be seen in the table:
>
> - (1) The majority of error types in the initial PoT responses are intrinsic reasoning errors (77.4%), such as misunderstanding the question or omitting conditions. The initial responses also exhibit a relatively high proportion (16.8%) of unreasonable output errors, while syntax and runtime errors are less frequent but not absent (5.8%).
> - (2) CRITIC has a high success rate in correcting unreasonable output and syntax errors (57.4% and 35.3% respectively). However, the correction rate for intrinsic errors, for which reliable feedback cannot be obtained, is relatively low (26.7%). Overall, CRITIC reduces errors in the initial erroneous samples by 32.2% in a non-oracle setting.
> - (3) Notably, while CRITIC has corrected a substantial number of errors in the initial PoT, as can be seen from the last row of the table above, there is a decrease of -4.3% in the accuracy of CRITIC on originally correct outputs. This results in the error modes after tool feedback also including 14.3% wrong corrections.
>
> We will incorporate these results into the paper in subsequent updates.
>
> ## Discussing The Role of Each Tool in Different Tasks
>
> > It is unclear how important each Tool is to each task. Such analysis will provide further insight into where the improvements come from.
>
> Thank you for your suggestion! We will incorporate an analysis of the significance of different tools in various scenarios within the CRITIC paper.
>
> To be specific, e.g., in knowledge-intensive tasks such as commonsense QA (AmbigNQ and TriviaQA) and multi-hop knowledge reasoning tasks like HotpotQA, the leading roles are played by web tools, and CRITIC primarily utilizes Wikipedia page browsing and Google snippet, as demonstrated by the numerous case studies in Appendix E.1; for mathematical program synthesis tasks, external knowledge is usually not required, and a code interpreter can serve the same function as a calculator. Therefore, in these experiments, our external feedback comes from error messages and execution results from the interpreter, as shown in the cases in Appendix E.2.
>
>
> ## Clarifying Sampling Config Variations
>
> > The authors used different sampling config for the experiments for different tasks: e.g. p=0.9 was used for section 4.3 which is different from p=0.5 in 4.1 and 4.2.
>
> CRITIC have applied p=0.9 in the toxicity reduction task, diverging from the p=0.5 used in other tasks. This decision is based on the standard set in the RealToxicityPrompts benchmark [2], and is consistent with the methodology employed across all baseline studies [3], ensuring fair comparison.
>
> ### References
>
> [1] Shao, Zhihong, et al. "Enhancing Retrieval-Augmented Large Language Models with Iterative Retrieval-Generation Synergy." arXiv preprint arXiv:2305.15294 (EMNLP 2023).
>
> [2] Gehman, Samuel, et al. "Realtoxicityprompts: Evaluating neural toxic degeneration in language models." arXiv preprint arXiv:2009.11462 (2020).
>
> [3] Welleck, Sean, et al. "Generating sequences by learning to self-correct." arXiv preprint arXiv:2211.00053 (2022).

---

> ### Comment · Reviewer_rWYL · 2023-11-19
>
> Thank you authors for adding the detailed error analysis on QA tasks, and my concerns are fully addressed.

---

> > ### Author Response · Authors · 2023-11-22
> > **Thank you!**
> >
> > Dear Reviewer rWYL,
> >
> > Thanks for your support! We are very happy that our revision and response addressed your concerns, and thank you again for your constructive feedback! And we wish you a Happy Thanksgiving!
> >
> > Best Regards,
> > The Authors

---

### Official Review · Reviewer_dDfr · 2023-11-01

**Soundness:** 3 good
**Presentation:** 3 good
**Contribution:** 3 good
**Rating:** 8
**Confidence:** 4

**Summary:**

The paper proposes a framework called CRITIC that enables large language models (LLMs) to self-verify and self-correct their outputs by interacting with external tools. The authors demonstrate the effectiveness of CRITIC in improving the performance of LLMs across multiple tasks, including free-form question answering, mathematical program synthesis, and toxicity reduction. The paper highlights the importance of external feedback in promoting the ongoing self-improvement of LLMs.

**Strengths:**

1) The paper introduces a novel framework, CRITIC, which addresses the limitations of LLMs by allowing them to verify and correct their outputs through interaction with external tools.

2) The authors provide comprehensive evaluations of CRITIC on different tasks and datasets, demonstrating its consistent performance improvement over baseline methods.

3) The paper highlights the crucial role of external feedback in the self-improvement of LLMs and emphasizes the unreliability of LLMs in self-verification.

**Weaknesses:**

1) I think this is a good paper. The motivation is strong: utilizing external feedback to enhance the model's ability. However, some recent studies [1] reported that large language models cannot self-correct themselves.  I acknowledge that [1] did not involve external tools, which is different from CRITIC's setting and it is a paper after CRITIC which is not necessarily be included, but it would be more comprehensive to include a discussion with these new studies in such a fast-moving field.

2) How much of the additional costs? Since calling external tools costs money. The authors should report the cost for each experiment.

3) In Appendix C.2, an important work active-prompt [2] should be included, which applies uncertainty estimation to chain-of-thought prompting.

[1] Large Language Models Cannot Self-Correct Reasoning Yet
[2] Active Prompting with Chain-of-Thought for Large Language Models

**Questions:**

How much of the additional costs? Since calling external tools costs money. The authors should report the cost for each experiment.

---

> ### Author Response · Authors · 2023-11-13
> **Author Response to Official Review by Reviewer dDfr**
>
> Dear Reviewer dDfr,
>
> Thank you for your comprehensive and detailed review of our paper! We sincerely appreciate your thorough and well-considered suggestions, which will help us make our work stronger! We're grateful for your recommendation to accept our paper!
>
> ### **Including a Discussion with Follow-up Works**
>
> > I think this is a good paper. The motivation is strong: utilizing external feedback to enhance the model's ability. However, some recent studies [1] reported that large language models cannot self-correct themselves. I acknowledge that [1] did not involve external tools, which is different from CRITIC's setting and it is a paper after CRITIC which is not necessarily be included, but it would be more comprehensive to include a discussion with these new studies in such a fast-moving field.
>
> Sure and thanks for your nice words! There are numerous recent studies, such as [1] that you mentioned. It is actually a follow-up to CRITIC and it kindly cites the CRITIC paper numerous times. These papers further provide additional experiments on various tasks like reasoning [1],  graph coloring[2], and planning [3], further validating the findings of unreliable self-correction of LLMs from CRITIC and providing further insights [1,2,3]. Thank you for your very thoughtful suggestion, and we will include discussions on these new studies in our subsequent updates.
>
> ### **Discussing Tool-Use Cost for Each Experiment**
>
> > How much of the additional costs? Since calling external tools costs money. The authors should report the cost for each experiment.
>
> Nice suggestion! We’ll include a new paragraph discussing the cost of tool use in the Appendix, which is actually all free!
>
> - For QA tasks, as mentioned in the footnote in Sec. 4.1, we build a Google Web Crawler to crawl the results of Google Search and web pages, and employ a caching mechanism for web search, storing about 9GB of search results from January to April 2023 during our experiments. This part of the code is separately open-sourced anonymously at https://anonymous.4open.science/r/llm-agent-web-tools. The results of the Search Engine in the paper are all obtained using this code. In addition, we will also open-source all caches after the anonymous review period ends, to ensure stability, fairness, and reproducibility in our results.
> - For Mathematical program synthesis tasks, we use a local code interpreter, which is free of charge.
> - For toxicity reduction tasks, we adopt PERSPECTIVE API [4] kindly provided by Google, which is also free.
>
> ### **Adding Active-Prompt to Uncertainty Estimation**
>
> > In Appendix C.2, an important work active-prompt [2] should be included, which applies uncertainty estimation to chain-of-thought prompting.
>
> Thank you for your suggestion!
> We concur that Active-Prompt is a successful application of uncertainty estimation in the context of demonstration selection. In response to your request, we have referenced this paper in Appendix C.2 in our paper.
>
> In light of these clarifications and revisions, we kindly request that you consider increasing the review score.
>
> ### References
>
> [1] Huang, Jie, et al. "Large language models cannot self-correct reasoning yet." arXiv preprint arXiv:2310.01798 (2023).
>
> [2] Stechly, Kaya, Matthew Marquez, and Subbarao Kambhampati. "GPT-4 Doesn't Know It's Wrong: An Analysis of Iterative Prompting for Reasoning Problems." arXiv preprint arXiv:2310.12397 (2023).
>
> [3] Valmeekam, Karthik, Matthew Marquez, and Subbarao Kambhampati. "Can Large Language Models Really Improve by Self-critiquing Their Own Plans?." arXiv preprint arXiv:2310.08118 (2023).
>
> [4] https://www.perspectiveapi.com/

---

> > ### Author Response · Authors · 2023-11-22
> >
> > Dear Reviewer dDfr, we greatly appreciate the your time in reviewing our response. Having submitted the rebuttal and revised our paper, we are eager to know if you have any remaining concerns. Since the reviewer-author discussion phase will end soon, we would greatly appreciate your support and valuable feedback before the deadline at Nov. 22.

---

> > > ### Comment · Reviewer_dDfr · 2023-12-02
> > >
> > > Dear authors,
> > >
> > > Thanks for your response. I have raised my score because my concerns have been well addressed. Thanks

---

### Official Review · Reviewer_ASEM · 2023-11-04

**Soundness:** 2 fair
**Presentation:** 2 fair
**Contribution:** 2 fair
**Rating:** 5
**Confidence:** 4

**Summary:**

This paper introduces CRITIC, a method for improving  the outputs of language models by leveraging external feedback from various tools. The idea is to generate an initial output with the language model and then refine this output using feedback from an external tool, such as a Python interpreter, search engines, or toxicity detection APIs. Notably, this approach relies solely on in-context learning without the need for specialized training. Results across various tasks, including question answering, mathematical reasoning, and toxicity reduction, show that CRITIC improves over baselines.

**Strengths:**

- The experiments demonstrate the effectiveness of the proposed approach across a diverse set of tasks, indicating its potential to significantly improve the performance of large language models (LLMs).

- Utilizing external feedback as a means of improving LLM outputs is practical. The simplicity of the approach is a plus, as it facilitates widespread application.

**Weaknesses:**

- The primary concern with this work is its novelty. Several studies have previously demonstrated that external feedback can be instrumental in correcting LLM outputs. In fact, there is existing work within each domain addressed in this paper, such as Self-Correct ([1], using external APIs), Self-Ask ([2], employing a search engine), and Self-Debug ([3], via a Python interpreter). Notably, Self-Debug and Self-Ask have a striking resemblance to CRITIC but are not referenced.


- The settings that rely on an oracle are somewhat idealistic, and detract from the core message of the paper. It may be more appropriate to move these results to an appendix (as done by other works) to facilitate a clearer understanding.



[1] Welleck, Sean, Ximing Lu, Peter West, Faeze Brahman, Tianxiao Shen, Daniel Khashabi, and Yejin Choi. "Generating Sequences by Learning to Self-Correct." In The Eleventh International Conference on Learning Representations. 2022.

[2] Press, Ofir, Muru Zhang, Sewon Min, Ludwig Schmidt, Noah A. Smith, and Mike Lewis. "Measuring and narrowing the compositionality gap in language models." arXiv preprint arXiv:2210.03350 (2022).


[3] Chen, Xinyun, Maxwell Lin, Nathanael Schärli, and Denny Zhou. "Teaching large language models to self-debug." arXiv preprint arXiv:2304.05128 (2023).

**Questions:**

- The emphasis in Table 1 seems inconsistent. For instance, the AmbigNQ EM score of 50.0 is highlighted for Text-Davinci-003, but it is not the highest. Is this a bug or am I missing something?

- Regarding the GSM task in a non-oracle setting, it appears that feedback from the interpreter is limited to syntactic correctness. Given the improvements, it suggests that many of the programs were initially syntactically wrong. Is this the case?

---

> ### Author Response · Authors · 2023-11-13
> **Author Response to Official Review by Reviewer ASEM (1/3)**
>
> Dear Reviewer ASEM,
>
> Thank you for your comprehensive and detailed review of our paper!
>
> We sincerely appreciate your thorough and well-considered suggestions, which will help us make our work stronger.
>
> ## Novelty and Contributions in Comparison to Previous and Contemporary Works
>
> >  Several studies have previously demonstrated that external feedback can be instrumental in correcting LLM outputs. In fact, there is existing work within each domain addressed in this paper, such as Self-Correct ([1], using external APIs), Self-Ask ([2], employing a search engine), and Self-Debug ([3], via a Python interpreter). Notably, Self-Debug and Self-Ask have a striking resemblance to CRITIC but are not referenced.
>
> Thank you for raising these concerns! We agree that there are numerous related works to CRITIC, given the importance and rapid development of LLM self-correction and tool-use topics. However, we would like to ****address the misconceptions and misunderstandings**** about CRITIC and its contributions when compared to many related works. This is because ****CRITIC's stance and findings significantly differ from these works****:
>
> ### **Intrinsic Self-Verification and Self-Correction**
>
> These works include Self-Critique [6], CAI [7], Reflexion[1], Self-Refine [2], and others [8-10], they prompt or train language models to correct their results. In contrast, our study is the first to demonstrate that such a "Self-Verification and Self-Correction" approach has proven to be **remarkably unreliable across diverse tasks and various LLMs**. Specifically, modest or even deteriorated performance is observed universally using self-correct without external feedback in Sec. 4 and Appendix D.1. Consequently, CRITIC emphasizes the importance of feedback from external interactions for the consistent self-improvement of LLMs. The proposed framework is general and has proven effective across multiple tasks. These profound reflections on unreliable self-correction of LLMs and crucial findings on the importance of external feedback can be important learnings for the community, as recognized by Reviewer [rWYL].
>
> - **Self-Correct** [11] uses the PERSPECTIVE API to build toxicity reduction pairs for training a corrector. Despite this, **its focus isn't on tool feedback but on creating value-improving pairs to train an additional corrector.** It doesn't use external feedback in math reasoning or constraint generation, primarily employing an intrinsic self-correction mechanism. It is mainly designed for smaller models like GPT-2, which have limited in-context learning capabilities. In contrast, our approach, CRITIC, iteratively optimizes outputs based on various external tool feedback, rather than solely relying on the model's own judgment. Additionally, CRITIC is a plug-and-play approach applicable to black-box LLMs, which further distinguishes it. We also provide an analysis and comparison to Self-Correct in our original paper's related works section and Sec 4.3's experiments.
> - You also referenced **Self-Debug** [3], which notably, debuted concurrently with the first version of CRITIC. Self-Debug emphasizes code tasks, proposing self-explanation **“without any feedback on code correctness or error messages” (intrinsic feedback)**, and applying unit tests similar to LEVER [13]. In contrast, results from CRITIC show that self-correct without external feedback is unreliable, and CRITIC addresses general challenges of truthfulness, trustworthiness, and toxicity, using various tools like Google and external APIs. In light of your valuable feedback, we'll include a comparison with Self-Debug in our updated paper.
>
> ### **Tool-Use: Tool Augmented LMs**
>
> Another category of very related work is Tool Augmented LMs, such as ReAct [4], PoT [5], and Toolformer [17]. We would like to clarify that **these works significantly differ from CRITIC as they focus on tool learning**. To the best of our knowledge, none of them consider using tool-interaction as faithful feedback to iteratively improve the LLMs, which is the key component of CRITIC.
>
> - You also mentioned a "striking resemblance" to the **Self-Ask** paper, which to our understanding, also does not involve self-correction with external feedback. In fact, Self-Ask “uses a search engine to answer sub-questions” and does not focus on verifying and correcting answers. However, we believe this work falls within the broader scope of RAG and will be referenced in the related works section.

---

> ### Author Response · Authors · 2023-11-13
> **Author Response to Official Review by Reviewer ASEM (2/3)**
>
> ### **Dive into The Unreliability of Self-correction**
>
> CRITIC further delves into the core reason behind the unreliability of self-verification and self-correction from the perspective of uncertainty estimation, as shown in Appendix D.1,. Essentially, language models are ****incapable of accurately identifying "what they know"**** (i.e., LLMs don't know what they know) [12] without relying on external tools. Therefore, without the aid of oracle verification (employed in many contemporary works such as Reflexion [1] and Self-Refine [2]), self-correction might surprisingly deteriorate performance for many tasks, and even incorrectly modify many initial answers (as demonstrated in Table 2, 3 under CRITIC w/o Tool, and in Table 8 under Self-Refine).
>
> - As noted by Reviewer [dDfr], a nice recent follow-up work to CRITIC, titled "LLMs cannot self-correct reasoning yet" [14] extends the study of CRITIC (which cites CRITIC numerous times). It further validates and expands our findings on the unreliability of Self-Verify and Self-Correct in CRITIC using GPT-4 in more settings on many reasoning tasks.
> - Another latest research [15, 16] experiments on graph coloring [15] and planing [16], demonstrating that "LLMs are in fact very poor at verifying solutions" and that "iterative prompting can help when there is an external provably correct verifier in the loop". These follow-ups further illustrates the significance and foresight of CRITIC's findings on the unreliability of Self-Verification and Self-Correction and the value of introducing external tool verification.
>
> In conclusion, CRITIC is the first to unveil the unreliability of self-verification and self-correction across diverse tasks and LLMs of various families and sizes. By initially highlighting the challenges LLMs encounter in self-verification [12], self-correction [6,7], our goal is to rectify any potential overestimations of these LLM's abilities within the research community [1-3]. By emphasizing that feedback from external tool interaction is crucial for consistent self-improvement of LLMs, we hope our findings provide valuable insights and encourage further exploration and enhancement of self-improving LLMs.
>
> Moreover, we have discussed these related works and offered a thorough comparison to the latest research in Appendix C. We will also expand the discussion to include recent follow-up works [18], such as the “LLMs cannot self-correct reasoning” paper [14] as suggested by reviewer [dDfr].
>
> ## Adjusting the Placement of Oracle Settings for Enhanced Clarity
>
> > The settings that rely on an oracle are somewhat idealistic, and detract from the core message of the paper. It may be more appropriate to move these results to an appendix (as done by other works) to facilitate a clearer understanding.
>
> In fact, the oracle setting in CRITIC was established and included in the main text with reference to previous work (see Table 1 in [11]). It's worth mentioning that contemporaneous works to CRITIC such as Reflexion [1], Self-Refine [2], and RCI [19] *all* used the Oracle setting (i.e., trial-and-error) for their main results. Since you suggest that it may be more appropriate to move these results, we will adjust the placement of the Oracle setting in subsequent updates, hoping this will make things clearer for you.
>
> ## Addressing Table 1 Emphasis Typos (Q1)
>
> > The emphasis in Table 1 seems inconsistent. For instance, the AmbigNQ EM score of 50.0 is highlighted for Text-Davinci-003, but it is not the highest. Is this a bug or am I missing something?
>
> Thank you for pointing out this nuance in Table 1. We acknowledge the discrepancy in highlighting the AmbigNQ scores, despite it not being the highest. This was indeed an oversight and not a bug. We have corrected this in an updated version. Additionally, we have meticulously reviewed all the tables to ensure no further inconsistencies are present. We appreciate your rigorous attitude and constructive feedback.

---

> ### Author Response · Authors · 2023-11-13
> **Author Response to Official Review by Reviewer ASEM (3/3)**
>
> ## Adding Additional Error Analysis on the Role of Interpreter Feedback (Q2)
>
> > Regarding the GSM task in a non-oracle setting, it appears that feedback from the interpreter is limited to syntactic correctness. Given the improvements, it suggests that many of the programs were initially syntactically wrong. Is this the case?
>
> Thanks for raising this issue!
> The feedback from the interpreter is not solely about syntactic correctness. As mentioned in Sec. 4.2 and demonstrated in the cases in Appendix E.2, CRITIC leverages both "error messages and execution results", enabling it to fix syntax errors, correct runtime errors (e.g., timeout error in Listing 9 in Appendix E.2) and self-reflect on unreasonable reasoning steps and outputs (e.g., Listing 7 in Appendix E.2). Notably, in non-oracle settings, we do not only apply correction on answers where the interpreter fails, but by default, we allow LLMs to verify and then correct if necessary and "stop if the executed result remains unchanged for two consecutive revisions", as detailed in Sec. 4.2.
>
> Moreover, to offer readers a more comprehensive understanding of the specific corrections made by CRITIC and the specific benefits derived from tool feedback, we carried out a manual statistical analysis of the types of corrections made by CRITIC on the GSM8k full test set (1319 samples).
>
> Specifically, we identified four different categories of initial program errors: syntax errors, runtime errors, unreasonable outputs (such as irrational negative values), and other intrinsic reasoning errors. We calculated the accuracy of the initial PoT (Init), and CRITIC for each type of error. The settings for corrections are consistent with the non-oracle setting in the original paper, with up to four rounds of correction. The statistics are presented in the following table:
>
> | Error Type          | Init (Count) | Init (Acc) | CRITIC (Count)| CRITIC (Acc)|
> |---------------------|-------------|----------|-------------|----------|
> | Intrinsic Error     | 281 (77.4%) | 0.0      | 206 (83.7%) | 26.7 |
> | Unreasonable Output | 61 (16.8%)  | 0.0      | 26 (10.6%)  | 57.4 |
> | Syntax Error        | 17 (4.7%)   | 0.0      | 11 (4.5%)   | 35.3 |
> | Runtime Error       | 4 (1.1%)    | 0.0      | 3  (1.2%)   | 25.0 |
> | All Init Errors     | 363         | 0.0      | 246         | 32.2 |
>
> As shown in the above table:
>
> - (1) The majority of error types in the initial PoT responses are intrinsic reasoning errors (77.4%), such as misunderstanding the question or omitting conditions. The initial responses also exhibit a relatively high proportion (16.8%) of unreasonable output errors, while syntax and runtime errors are less frequent but not absent (5.8%).
> - (2) CRITIC has a high success rate in correcting unreasonable output and syntax errors (57.4% and 35.3% respectively). However, the correction rate for intrinsic errors, for which reliable feedback cannot be obtained, is relatively low (26.7%). Overall, CRITIC reduces errors in the initial erroneous samples by 32.2% in a non-oracle setting.
>
> We hope our responses adequately addresses all your concerns! We greatly appreciate your time and effort in reviewing our paper, and your constructive feedback!

---

> ### Author Response · Authors · 2023-11-13
> **Author Response to Official Review by Reviewer ASEM (References)**
>
> ### References
>
> [1] Shinn, Noah, et al. "Reflexion: Language agents with verbal reinforcement learning." Thirty-seventh Conference on Neural Information Processing Systems. 2023.
>
> [2] Madaan, Aman, et al. "Self-refine: Iterative refinement with self-feedback." Thirty-seventh Conference on Neural Information Processing Systems. 2023.
>
> [3] Chen, Xinyun, et al. "Teaching large language models to self-debug." arXiv preprint arXiv:2304.05128 (2023).
>
> [4] Yao, Shunyu, et al. "React: Synergizing reasoning and acting in language models." arXiv preprint arXiv:2210.03629 (2022).
>
> [5] Chen, Wenhu, et al. "Program of thoughts prompting: Disentangling computation from reasoning for numerical reasoning tasks." arXiv preprint arXiv:2211.12588 (2022).
>
> [6] Saunders, William, et al. "Self-critiquing models for assisting human evaluators." arXiv preprint arXiv:2206.05802 (2022).
>
> [7] Bai, Yuntao, et al. "Constitutional ai: Harmlessness from ai feedback." arXiv preprint arXiv:2212.08073 (2022).
>
> [8] Campos, Jon Ander, and Jun Shern. "Training language models with language feedback." ACL Workshop on Learning with Natural Language Supervision. 2022.. 2022.
>
> [9] Chen, Angelica, et al. "Improving code generation by training with natural language feedback." arXiv preprint arXiv:2303.16749 (2023).
>
> [10] Haluptzok, Patrick, Matthew Bowers, and Adam Tauman Kalai. "Language models can teach themselves to program better." *arXiv preprint arXiv:2207.14502* (2022).
>
> [11] Welleck, Sean, et al. "Generating Sequences by Learning to Self-Correct." *The Eleventh International Conference on Learning Representations*. 2023.
>
> [12] Kadavath, Saurav, et al. "Language models (mostly) know what they know." arXiv preprint arXiv:2207.05221 (2022).
>
> [13] Ni, Ansong, et al. "Lever: Learning to verify language-to-code generation with execution." International Conference on Machine Learning. PMLR, 2023.
>
> [14] Huang, Jie, et al. "Large language models cannot self-correct reasoning yet." arXiv preprint arXiv:2310.01798 (2023).
>
> [15] Stechly, Kaya, Matthew Marquez, and Subbarao Kambhampati. "GPT-4 Doesn't Know It's Wrong: An Analysis of Iterative Prompting for Reasoning Problems." arXiv preprint arXiv:2310.12397 (2023).
>
> [16] Valmeekam, Karthik, Matthew Marquez, and Subbarao Kambhampati. "Can Large Language Models Really Improve by Self-critiquing Their Own Plans?." arXiv preprint arXiv:2310.08118 (2023).
>
> [17] Schick, Timo, et al. "Toolformer: Language models can teach themselves to use tools." arXiv preprint arXiv:2302.04761 (2023).
>
> [18] Pan, Liangming, et al. "Automatically correcting large language models: Surveying the landscape of diverse self-correction strategies." *arXiv preprint arXiv:2308.03188* (2023).
>
> [19] Kim, Geunwoo,, et al. "Language models can solve computer tasks." Thirty-seventh Conference on Neural Information Processing Systems. 2023.

---

> ### Author Response · Authors · 2023-11-21
>
> Dear Reviewer ASEM, We hope that our revised paper and the above responses address your concerns. If there are any unresolved issues, please let us know. We are more than happy to answer any further questions you may have.

---

> ### Comment · Reviewer_ASEM · 2023-11-21
> **Thanks for your response! Some clarifications**
>
> Thanks for taking the time to respond! Let me reiterate my concerns and also rephrase my understanding of your responses.
>
> ---
>
> ### Weakness 1: Novelty
>
> > The primary concern with this work is its novelty. Several studies have previously demonstrated that external feedback can be instrumental in correcting LLM outputs. In fact, there is existing work within each domain addressed in this paper, such as Self-Correct ([1], using external APIs), Self-Ask ([2], employing a search engine), and Self-Debug ([3], via a Python interpreter). Notably, Self-Debug and Self-Ask have a striking resemblance to CRITIC but are not referenced.
>
>
>
> 1. Self-Correct
>
> > "its focus isn't on tool feedback but on creating value-improving pairs to train an additional corrector."
>
> This doesn't seem to be correct?
>
> From [self-correct](https://arxiv.org/pdf/2211.00053.pdf): *As the value function, we use the Perspective API score, v(y) ∈ [0, 1], which measures the toxicity of the completed sequence* They use the word "value function," but essentially use the score from the API to decide if the improvements have any benefit.
>
> Further:
>
> _We use additional fine-grained information from the toxicity API as natural language
> feedback. Specifically, besides the overall toxicity score, Perspective API also provides scores for
> fine-grained attributes of toxicity (e.g., identity attack, profanity, flirtation, etc.). At training time,
> we compare the attribute scores from a hypothesis and its selected correction, and use the attribute
> with the largest decrease as natural language feedback (e.g. "decrease toxicity in profanity"). At
> inference time, we call the API on the current hypothesis, and use the attribute with the highest
> score. Here **we use the API at inference time**, which is not required in our previous experiments._
>
>
>
> 2. Self-Debug:
>
>
> > Self-Debug emphasizes code tasks, proposing self-explanation "without any feedback on code correctness or error messages" (intrinsic feedback)
>
> This also doesn't seem to be precise. From [Self-Debug v1](https://arxiv.org/pdf/2304.05128v1.pdf): Figure 1: *The **code explanation** along with the execution results constitute the feedback message, which is then sent back to the model to perform more debugging steps...*. Please also see Section 3 and the examples in the Appendix.
>
> So, I wonder if the novelty claim holds.
>
> *Regarding concurrency with Self-Debug:* Self-Debug came out on April 11 2023, so as per ICLR reviewer guidelines it is not concurrent work (only work after posted after **May 28 2023** is concurrent). Please see the [guidelines](https://iclr.cc/Conferences/2024/ReviewerGuide) here. I am not aware when CRITIC came out, and I do not want to look it up either. I'll let the AC decide on this.
>
> ---
>
> ###  Weakness 2: Including Oracle results in the main paper
>
> > The settings that rely on an oracle are somewhat idealistic,
>
> Thanks for moving the Oracle results to the Appendix.
>
> > It's worth mentioning that contemporaneous works to CRITIC such as Reflexion [1], Self-Refine [2], and RCI [19] all used the Oracle setting (i.e., trial-and-error) for their main results.
>
> This is incorrect, at least for some of the works. For example, [2] ([link](https://arxiv.org/pdf/2303.17651.pdf)) place the results from the Oracle setting in the Appendix. On this note,
>
> >  CRITIC is the first to unveil the unreliability of self-verification and self-correction across diverse tasks and LLMs of various families and sizes.
>
> This statement may be toned down because [2] in the version posted on May 25th includes a discussion on this on Page 5 and also presents initial conjectures for why models may fail to reason (lack of good quality feedback)
>
>
> Nevertheless, this weakness is addressed by moving the results to the Appendix.
>
> ---
>
> ### Question: Nature of feedback from the interpreter
>
> Thanks for the analysis!
>
> 1. How is "Unreasonable Output" determined? Is it a hard-coded rule that the output cannot be negative or an irrational number?
>
> 2. From the analysis: *The majority of error types in the initial PoT responses are intrinsic reasoning errors (77.4%), such as misunderstanding the question or omitting conditions.* To clarify, there is no way to catch these errors with tools, right?

---

> ### Author Response · Authors · 2023-11-22
> **Thank you for your reply!**
>
> Thank you very much for your quick reply and for letting us know your remaining concerns!
>
> ### Further Discussion on Novelty
>
> > **1. Self-Correct**
> >
> > > "its focus isn't on tool feedback but on creating value-improving pairs to train an additional corrector."
> >
> > This doesn't seem to be correct?
> >
> > From self-correct: *As the value function, we use the Perspective API score, v(y) ∈ [0, 1], which measures the toxicity of the completed sequence* They use the word "value function," but essentially use the score from the API to decide if the improvements have any benefit.
> >
>
> We agree that Self-Correct uses the Perspective API in one of its tasks, but we maintain our assertion because it does not use external feedback in math reasoning or constraint generation, primarily employing an intrinsic self-correction mechanism. The focus of this paper is not on feedback based on tool-interaction, but on "creating value-improving pairs to train an additional corrector.”
>
> Unlike Self-Correct, we have validated the effectiveness of self-correction with external tool-interaction on diverse tasks and LLMs through extensive experiments. Moreover, we have demonstrated that self-correction based on external feedback does not require retraining an additional corrector and can be taught with few-shot in-context learning, making it applicable to black-box LLMs.  Furthermore, we provide an analysis and comparison to Self-Correct in our original paper's related works section and Section 4.3's experiments.
>
> > **2. Self-Debug:**
> > > Self-Debug emphasizes code tasks, proposing self-explanation "without any feedback on code correctness or error messages" (intrinsic feedback)
> >
> > This also doesn't seem to be precise. From Self-Debug v1: Figure 1: *The **code explanation** along with the execution results constitute the feedback message, which is then sent back to the model to perform more debugging steps...*. Please also see Section 3 and the examples in the Appendix.
> >
> > So, I wonder if the novelty claim holds.
> >
>
> We clarify that this statement is from the abstract of [Self-Debug v1](https://arxiv.org/pdf/2304.05128v1.pdf) [2], which you also referenced. Thanks for you comment, we find that, despite Self-Debug focusing on code tasks and CRITIC concentrating on math reasoning program synthesis, we find no substantial difference in their core method of using program execution results as feedback.
>
> However, it's crucial to note that CRITIC was independently developed from January to April 2023, separate from Self-Debug. In fact, **Self-Debug and CRITIC were published within approximately a month** of each other, and Self-Debug is also [under review at ICLR’24](https://openreview.net/forum?id=KuPixIqPiq). While Self-Debug focuses on code tasks, CRITIC tackles broader challenges of truthfulness, trustworthiness, and toxicity, utilizing various tools such as Google and external APIs. Through extensive experimentation, we've demonstrated significant performance improvements offered by CRITIC across different base LLMs in tasks like QA, Math reasoning, and toxicity reduction. In light of your feedback, we will include a citation to this paper in our revision.

---

> > ### Author Response · Authors · 2023-11-22
> >
> > ### Including Oracle results in the main paper
> >
> > > > It's worth mentioning that contemporaneous works to CRITIC such as Reflexion [1], Self-Refine [2], and RCI [19] all used the Oracle setting (i.e., trial-and-error) for their main results.
> > >
> > > This is incorrect, at least for some of the works. For example, [2] ([link](https://arxiv.org/pdf/2303.17651.pdf)) place the results from the Oracle setting in the Appendix. On this note,
> > >
> > > > CRITIC is the first to unveil the unreliability of self-verification and self-correction across diverse tasks and LLMs of various families and sizes.
> > >
> > > This statement may be toned down because [2] in the version posted on May 25th includes a discussion on this on Page 5 and also presents initial conjectures for why models may fail to reason (lack of good quality feedback)
> > >
> > > Nevertheless, this weakness is addressed by moving the results to the Appendix.
> > >
> >
> > Thank you for acknowledging the concern addressed, but we need to clarify some potential misunderstandings.
> >
> > It's indeed correct that Self-Correct [1], Reflexion [3], RCI [4] use Oracle in their main results. Furthermore, the first version of Self-Refine ([link](https://arxiv.org/pdf/2303.17651v1.pdf)) reported Oracle results for math reasoning, and this is also verifiable through [its code](https://github.com/madaan/self-refine/blob/1a46d0c5a5ae600577cb84bc1193ac8504ca83a6/src/gsm/gsm_selfref_eval.py#L75-L93).
> >
> > We noted that after CRITIC's release, Self-Refine moved its oracle results to the appendix and includes the discussion you mentioned in its updated version on May 25th. **We cannot confirm whether the authors of Self-Refine read CRITIC before these modifications, but we do find these added discussions you mentioned were post-CRITIC**. This affirms our statement that "CRITIC is the first to unveil the unreliability of self-verification and self-correction across diverse tasks and LLMs of various families and sizes.”
> >
> > ### Answer to the Question: Nature of feedback from the interpreter
> >
> > > How is "Unreasonable Output" determined? Is it a hard-coded rule that the output cannot be negative or an irrational number?
> > >
> >
> > Indeed, our error categorization process combines both rules and manual inspection. Given the large volume of data, conducting a full manual analysis would be quite time-consuming. Hence, after manually analyzing the GSM8k data, we observed that almost all ground truths are positive. Therefore, we identified data with negative values as unreasonable. While negative values don't always signify unreasonableness, this rule was determined based on the data characteristics.
> >
> > Please note that this filtering process is solely for the convenience of error analysis, and it doesn't imply that CRITIC can only identify negative values through tool-interaction feedback. We have also observed that CRITIC can analyze unreasonable decimals or numbers that are inconsistent with the problem statement. However, these instances are relatively less common in the GSM8k dataset and are challenging to filter out simply.
> >
> > If you believe there are areas in our error analysis that could be improved, please don't hesitate to suggest them. We can make rapid improvements before the discussion deadline.
> >
> > > From the analysis: The majority of error types in the initial PoT responses are intrinsic reasoning errors (77.4%), such as misunderstanding the question or omitting conditions. To clarify, there is no way to catch these errors with tools, right?
> > >
> >
> > While a code interpreter executing the original program cannot directly catch these intrinsic errors, we should not overstate that "there is no way to catch these errors with tools." In fact, we can follow the CRITIC approach and leverage tool-use results for simulated reflection [6] or employ code-based self-verification [7] to partially identify and resolve these errors. This follow-up works further validates the promising direction of allowing LLMs to interact with tools for self-verification and correction, as proposed by CRITIC.
> >
> > We sincerely thank you for your thoughtful feedback and hope our response has addressed your concerns appropriately.
> >
> > ---
> >
> > ### References
> >
> > [1] Welleck, Sean, et al. "Generating sequences by learning to self-correct."
> >
> > [2] Chen, Xinyun, et al. "Teaching large language models to self-debug." https://openreview.net/forum?id=KuPixIqPiq
> >
> > [3] Shinn, Noah, et al. "Reflexion: Language agents with verbal reinforcement learning."
> >
> > [4] Kim, Geunwoo,, et al. "Language models can solve computer tasks."
> >
> > [5] Madaan, Aman, et al. "Self-refine: Iterative refinement with self-feedback."
> >
> > [6 ] Wang, Xingyao, et al. "Mint: Evaluating llms in multi-turn interaction with tools and language feedback." https://arxiv.org/pdf/2309.10691.pdf
> >
> > [ 7] Zhou, Aojun, et al. "Solving challenging math word problems using gpt-4 code interpreter with code-based self-verification." https://arxiv.org/pdf/2308.07921.pdf

---

> ### Author Response · Authors · 2023-11-22
>
> Dear Reviewer ASEM,
>
> Thank you once again for your review of our paper! We hope that our responses have adequately addressed all your concerns. As the deadline is approaching, we encourage you to reach out if you have any additional questions before the reviewer-author discussion period ends. We are more than happy to address any further questions or concerns you may have!
>
> Best Regards,
>
> The Authors

---

> ### Comment · Reviewer_ASEM · 2023-11-22
> **Thanks for your response**
>
> > We agree that Self-Correct uses the Perspective API in one of its tasks, but we maintain our assertion because it does not use external feedback in math reasoning or constraint generation, primarily employing an intrinsic self-correction mechanism.
>
> Essentially, Self-Correct did tool-driven correction for one task, and CRITIC does this for multiple tasks. I agree with this, but I don't know if this is grounds for novelty.
>
>
> > Thanks for you comment, we find that, despite Self-Debug focusing on code tasks and CRITIC concentrating on math reasoning program synthesis, we find no substantial difference in their core method of using program execution results as feedback.
>
> If I am reading this right, Self-Debug and CRITIC have the same method but focus on different tasks. I also agree with this statement.
>
> This discussion reinforces my concerns regarding novelty: the ideas in CRITIC are already present in Self-Correct and Self-Debug. However, I agree that the idea is extended to a broader range of tasks in CRITIC. I will consult with other reviewers during the remainder of the discussion period and try to understand why they think the idea is novel.
>
>
> Finally, regarding the dates and arxiv versions: my job is to judge the version submitted to ICLR, and my feedback is solely based on the submitted/revised version. I will let AC/SAC judge the arxiv version of CRITIC when they make a decision.
>
> Thanks

---

> > ### Author Response · Authors · 2023-11-23
> > **Summary and Appreciation to Reviewer ASEM**
> >
> > Dear Reviewer ASEM,
> >
> > We sincerely appreciate your active response and letting us know your thoughts!
> >
> > We are delighted to have had a vibrant discussion with you about CRITIC, especially its relationship with Self-Correct and Self-Debug. While we reached many agreements, we acknowledge a few unresolved debates. As we're unable to participate in further discussions due to procedural constraints, allow us to summarize our viewpoint and underscore CRITIC's contributions.
> >
> > We agree with your observation that before CRITIC, Self-Correct had already implemented "tool-driven correction" in a toxicity task (more precisely, by training additional correctors). Specifically, although it did not introduce tools in math or constrained generation tasks and relied solely on intrinsic feedback from the model, it indeed used the Perspective API to train a GPT-2 corrector in the toxicity task and also used the API at inference time. We completely agree with these facts.
> >
> > In fact, when we proposed CRITIC, we indeed took Self-Correct as an important reference work and used it as a baseline for analysis and comparison in our experiments (Sec. 4.3). Based on these consensuses, we emphasize the multi-faceted contributions of CRTIC as follows:
> >
> > 1. Firstly, we propose a unified CRITIC framework that integrates different tools and tasks into one framework, and design a series of new prompting methods that teach black-box LLMs to self-verify and self-correct through interaction with tools. This enables LLMs to have reliable self-correction capabilities through in-context learning, without relying on extra training (`as you summarized in the Review`).
> > 2. Secondly, we conduct comprehensive experiments across distinct tasks, demonstrating significant performance improvements offered by CRITIC across different LLMs and model sizes (`as acknowledged by you and all other reviewers`).
> > 3. Furthermore, we are the first to discover the unreliability of LLMs in self-correction through systematical experiments. We delve deeper into the main reason for the instability of Self-Correction from the perspective of uncertain estimation, finding that LLMs struggle to “know what they know” without external feedback ([Appendix D.1](https://openreview.net/pdf?id=Sx038qxjek)). Therefore, we emphasize that feedback from external tool interaction is crucial for the consistent self-improvement of LLMs.  (`In response to this contribution, you mentioned that Self-Refine proposed a similar viewpoint in its May revision. We pointed out that the revision of this paper was later than the publication of CRITIC and did not receive any subsequent opposition.`)
> >
> > Regarding Self-Debug, we reaffirm that CRITIC and Self-Debug are completely independent concurrent works published on ArXiv within a month of each other. We emphasize the unique contributions of CRITIC mentioned above. And we agree with your suggestion to leave the relationship between CRITIC and Self-Debug to the AC's discretion.
> >
> > Once again, we express our heartfelt appreciation for your meticulous review and constructive suggestions, which have significantly improved our paper. It's fair to say that you are one of the most responsible and proactive reviewers we have encountered in this submission process! We wish you a happy Thanksgiving!
> >
> > Wish you all the best,
> >
> > The Authors

---

### Author Response · Authors · 2023-11-14
**Author Response to all Reviewers**

# Author Response to all Reviewers

Firstly, we would like to extend our deepest gratitude to the reviewers for their comprehensive reviews and valuable feedback!

### **Positive Remarks**

We are glad to receive positive comments on our paper, including:

- The recognition of our research topic on Tool-Use and self-correction as being "very timely and important" [rWYL] and the appreciation of the strong motivation behind our work [dDfr].
- The acknowledgment of the merits of our proposed CRITIC method, which is seen as a valuable contribution [dDfr, rWYL], that is "practical and simple" [ASEM], with an "interesting comparison" [hutJ].
- The agreement on the significance of our paper in emphasizing the unreliability of LLMs in self-verification and self-correction [dDfr, rWYL], and the crucial role of external feedback [dDfr, rWYL], which is deemed as "an important learning for the community" [rWYL].
- The positive feedback on the clarity of our writing [dDfr, rWYL, hutJ] and our comprehensive experiments covering various downstream tasks and different model families and sizes [ASEM, dDfr, rWYL], which are found to show universal improvements [dDfr, rWYL].

Our intention with this paper is to unveil the unreliability of LLM self-verification [1] and self-correction [2, 3] across diverse tasks and LLMs of various sizes and families, and highlight external feedback. Our goal is to rectify potential overestimations of these LLM's abilities within the research community [4,5,6]. By emphasizing the importance of feedback from external tool interaction for the consistent self-improvement of LLMs, we aim to provide valuable insights that can stimulate further exploration and development in the field of self-improving LLMs!

### **Revisions Based on Feedback**

We also highly value the constructive suggestions and have made the following revisions to improve the clarity of our work:

- We have included a detailed error analysis for QA and Mathematical Program Synthesis tasks (Appendix D.2) [ASEM, rWYL].
- We have added a new discussion on the cost of tool-use for each experiment (Appendix D.3) and have open-sourced web tools at https://anonymous.4open.science/r/llm-agent-web-tools [dDfr].
- We have included a new discussion about the latest contemporary and follow-up works of CRITIC (Sec. 2 and Appendix C.1) [dDfr].
- We have discussed the role of each tool in different tasks (Appendix D.4) [rWYL].
- We have adjusted the placement of oracle settings for enhanced clarity, and addressed the Table 1 Emphasis Typos [ASEM], and the phrase issue regarding "mimic human" [hutJ].

We have responded to each reviewer individually and have made necessary revisions accordingly. We hope that our responses and revisions effectively address your concerns, and we would greatly appreciate it if you would consider revising your score!

### References

[1] Kadavath, Saurav, et al. "Language models (mostly) know what they know." arXiv preprint arXiv:2207.05221 (2022).

[2] Saunders, William, et al. "Self-critiquing models for assisting human evaluators." arXiv preprint arXiv:2206.05802 (2022).

[3] Bai, Yuntao, et al. "Constitutional ai: Harmlessness from ai feedback." arXiv preprint arXiv:2212.08073 (2022).

[4] Shinn, Noah, et al. "Reflexion: Language agents with verbal reinforcement learning." Thirty-seventh Conference on Neural Information Processing Systems. 2023.

[5] Madaan, Aman, et al. "Self-refine: Iterative refinement with self-feedback." Thirty-seventh Conference on Neural Information Processing Systems. 2023.

[6] Kim, Geunwoo,, et al. "Language models can solve computer tasks." Thirty-seventh Conference on Neural Information Processing Systems. 2023.

---

### Meta-Review · Area_Chair_Jcn5 · 2023-12-07

**Metareview:**

The authors introduce CRITIC, an approach aimed at improving the performance of LLMs by incorporating feedback from external tools. This method starts with generating a response using LLMs, which is then refined using input from external sources, including Python interpreters, search engines, or toxicity detection APIs. This refinement relies exclusively on in-context learning. The efficacy of CRITIC is demonstrated across various tasks, including question answering, mathematical problem-solving, and toxicity reduction.

Reviewers ASEM, dDfr, and hutJ have raised concerns regarding the novelty and differentiation from other works that utilize feedback from external tools. Nonetheless, considering the completeness and thoroughness of the work, I recommend an accept (poster) rating.

**Justification For Why Not Higher Score:**

As previously mentioned, several reviewers continue to express concerns regarding novelty.

**Justification For Why Not Lower Score:**

I appreciate the thoroughness of the study regarding the use of feedback from external tools for improving LLMs.

---

### Decision · Program_Chairs · 2024-01-16

Accept (poster)